# MaiT: Integrating Spatial Locality into Image Transformers with Attention Masks

## Abstract

Though image transformers have shown competitive results with convolutional neural networks in computer vision tasks, lacking inductive biases such as locality still poses problems in terms of model efficiency especially for embedded applications. In this work, we address this issue by introducing attention masks to incorporate spatial locality into self-attention heads. Local dependencies are captured with masked attention heads along with global dependencies captured by original unmasked attention heads. With Masked attention image Transformer – MaiT, top-1 accuracy increases by up to 2.5% compared to DeiT, without extra parameters/FLOPs, or external training data. Moreover, attention masks regulate the training of attention maps, which facilitates the convergence and improves the accuracy of deeper transformers. Masked attention heads guide the model to focus on local information in early layers and promote diverse attention maps in latter layers. Deep MaiT improves the top-1 accuracy by up to 1.7% compared to CaiT with fewer parameters and FLOPs. Encoding locality with attention masks requires no extra parameter or structural change, and thus it can be combined with other techniques for further improvement in vision transformers.

## 1 Introduction

Convolutional neural networks (CNNs) (Krizhevsky et al., 2012; He et al., 2016; Tan & Le, 2019) have been the de facto model for computer vision (CV) tasks, which are inherently equipped with inductive biases such as translation equivariance and locality. Recently, vision transformers (Dosovitskiy et al., 2021; Touvron et al., 2021a) are gaining momentum translating the success from transformer-based models in natural language processing (NLP) tasks (Vaswani et al., 2017; Devlin et al., 2019). Self-attention heads excel at capturing the long-range dependencies in sequences but struggle at focusing on local information. Unlike CNNs which have gone through many iterations of optimization, vision transformers are not very efficient and still require huge computing power and a large number of Flops.

Naturally, combining the benefits of both CNNs and vision transformers is promising to further boost performances of CV models. The question remains how to effectively integrate inductive bias such as spatial locality into transformers. One direction is to utilize convolutional blocks to extract spatial information by adapting either the patch-token embedding layer, self-attention module or feed-forward layers, to form CNN-transformer hybrid structures as in Li et al. (2021b); Srinivas et al. (2021); Graham et al. (2021); Wu et al. (2021a); Yuan et al. (2021a); Guo et al. (2021). However, forcefully inserting convolutional operations into transformers may potentially constrain the learning capacity of transformers.

To capture the spatial information without significantly changing the transformer model architecture, Chu et al. (2021b) introduce extra positional encoding. Han et al. (2021); Chen et al. (2021) fuse local and global representations using multiple transformer blocks or branches to simultaneously process images at different scales such as pixel-level, small-patch or large patch. Yuan et al. (2021b) apply a layerwise tokens-to-tokens transformation to capture local structure. These approaches usually come with the cost of extra parameters and model complexity, thus potentially lowering the inference speed.

d'Ascoli et al. (2021); Zhou et al. (2021) improve the self-attention for better representation with gated positional self-attention and learnable transformation matrix respectively. Hu et al. (2019),

Ramachandran et al. (2019) adapt the self-attention module and improve the performance of CNN models. Liu et al. (2021) captures locality within shifted windows in a hierarchical structure. Though it could save computation in some cases, the added model complexity may lower its potentials.

Different from prior works, we try to incorporate spatial locality without changing its architecture or adding extra parameters/FLOPs. We propose attention masks to guide the attention heads to focus on local information. Masked attention heads extract local dependencies more efficiently by allowing information aggregation only from the closest neighbors. This liberates other unmasked heads to learn global information more effectively. We name the modified model, Masked attention image Transformer (MaiT), which is built on top of DeiT (Touvron et al., 2021a). MaiT gathers both local and global information at the same time from different heads. Moreover, the regularization effects from attention masks facilitate the training of deep transformers by guiding the attention map learning and promoting diversity across transformer layers.

We proved that less is more in this specific case with attention masks. MaiT achieves up to 2.5% higher top-1 accuracy on ImageNet (Deng et al., 2009) with the same model architecture as DeiT (Touvron et al., 2021a). Additionally, deep MaiT outperforms CaiT (Touvron et al., 2021b) by up to 1.7% in top-1 accuracy with fewer parameters and simpler structure.

In summary, we make three major contributions in this work: 1). we propose attention masks to encode the spatial locality into self-attention heads without structural change or extra parameter/computation, while improving model efficiency. 2). We present a quick empirical search strategy for the masking scheme exploration, along with an automatic end-to-end search alternative. 3). We also reveal the importance of the locality across layers and the performance impact of attention masks on deep vision transformers. Note though MaiT is demonstrated with DeiT/CaiT, the attention mask is applicable to other vision transformers as well.

## 2 RELATED WORK

Spatial locality is an integral part of the convolutional operation with weight filters attending to local regions of input feature maps. Vision transformer (ViT) (Dosovitskiy et al., 2021) is the first pure transformer-based model on vision tasks, but it requires a large private labeled dataset JFT300M (Sun et al., 2017) to achieve competitive performances. Data-efficient image transformer (DeiT) (Touvron et al., 2021a) improves upon ViT models by introducing stronger data augmentation, regularization, and knowledge distillation. Class-attention in image transformer (CaiT) (Touvron et al., 2021b) extends DeiT by increasing the number of transformer layers. To overcome the difficulties of training deeper transformers, CaiT introduces LayerScale and class-attention layers, which increase the parameters and model complexity.

Tokens-to-Token vision transformer (T2T) (Yuan et al., 2021b) proposes an image transformation by recursive token aggregation to capture local structure. Stand-alone self-attention (Ramachandran et al., 2019) applies local self-attention layer to replace spatial convolution and outperform original ResNet models. Even though sharing value and key spatially is parameter efficient in this approach, content-based information is lost. Transformer-iN-Transformer (TNT) (Han et al., 2021) models both patch-level and pixel-level representations and applies outer and inner transformer blocks to extract global and local information respectively. ConViT (d'Ascoli et al., 2021) proposes the gated positional self-attention to incorporate soft convolutional biases. CrossViT (Chen et al., 2021) proposes a dual-branch transformer architecture for multi-scale feature extraction.

For pixel-level prediction tasks such as semantic segmentation, object detection, Pyramid Vision Transformer (PVT) (Wang et al., 2021a) introduces a progressive shrinking pyramid and spatial-reduction attention with fine-grained image patches. DETR (Carion et al., 2020) adapts transformers for object detection tasks. Swin Transformer (Liu et al., 2021) applies hierarchical transformer with shifted windows of varying sizes. Twins (Chu et al., 2021a) deploys interleaved locally-grouped self-attention and global sub-sample attention layers to improve performances.

To optimize transformer and save computation, Wu et al. (2021b) uses centroid attention to extract and compress input information, Jaegle et al. (2021) iteratively distill inputs into latent space with attention bottlenecks, Wang et al. (2021b) dynamically adjusts the number of tokens with multiple cascading transformers, and Wu et al. (2020) introduced semantic token to replace pixel-based transformers to save computation.

There are hybrid architectures fusing convolutional and transformer blocks, such as LocalViT (Li et al., 2021b), BoTNet (Srinivas et al., 2021), LeViT (Graham et al., 2021), BossNet (Li et al., 2021a), CvT (Wu et al., 2021a), CoaT (Xu et al., 2021), CMT (Guo et al., 2021) for higher accuracy and faster inference.

Unlike prior literature, our work explores the intrinsic capability of pure transformer block on incorporating spatial locality and the impact of the locality along the depth direction. This work is also inspired by the emerging graph attention network (Veličković et al., 2018), borrowing the concept of message passing and information aggregation from nearest neighbors.

Note attention masks have been used in NLP tasks as a sparsification method to reduce the computation complexity, as well as to capture local information, as in Guo et al. (2019); Child et al. (2019); Beltagy et al. (2020); Ainslie et al. (2020).

## 3 VISION TRANSFORMER PRELIMINARIES

The transformer architecture introduced by Vaswani (Vaswani et al., 2017) inspired many model variants with remarkable success in NLP tasks. ViT (Dosovitskiy et al., 2021) extends pure transformer-based architecture into CV applications. Instead of pixel-level processing, ViT splits the original images into a sequence of patches as inputs and transforms them into patch tokens, for better computation efficiency. In general, ViT consists of 3 fundamental modules: embedding layer, multi-head self-attention, and feed-forward network.

To process images in transformer, the original RGB images ($224 \times 224$) is flattened into a sequence of $N$ ($14 \times 14$) patches. Each patch has a fixed size (typical 16x16 pixels). Patches are then transformed into patch embedding with hidden dimensions ($D$) of 192, 384, and 768 for tiny, small, and base models respectively in ViT/DeiT. In addition to patch tokens, the embedding layer also integrates positional information, classification and knowledge distillation through the positional token, class token and distillation token, respectively.

Positional token is added into the patch embedding with a trainable positional embedding. However, this positional embedding is added only in the embedding layers. The spatial information is largely lost in the transformer layers since all-to-all attention is invariant to the order of the patches.

The class token is another trainable vector ($1 \times D$), concatenated to the patch tokens (total $N+1$). It is used to collect information from the patch tokens to make output predictions, while also spreading information among patch tokens during training.

Distillation token is sometimes added for knowledge transfer from teacher models, such as a CNN model. When training the distilled version of the model, a distillation token is further concatenated to the patch token along with the class token (total $N+2$).

Multi-head self-attention (MHA) module has multiple parallel attention heads, where each of them comprises three main components: Key ($K$), Query ($Q$), and Value ($V$), Key and Query are trained and multiplied to estimate how much weights on each corresponding token in Value for output:

$$Attention(K, Q, V) = Softmax\left(\frac{QK^T}{\sqrt{d}}\right) V \qquad (1)$$

Where softmax is applied to each row of the input product matrix ($QK^T$) and $\sqrt{d}$ provides appropriate normalization. Multiple attention heads in MHA attend to different parts of the input simultaneously. Considering $H$ heads in MHA layer, the hidden dimension $D$ is split equally across all heads ($D = H \times d$).

Feed-forward network (FFN) follows after the MHA module, containing two linear transformation layers separated by Gelu activation. The hidden dimension expands by 4x after the first linear layer from $D$ to $4D$, and is reduced back to $D$ in the second linear layer. Both MHA and FFN use skip-connections with layer normalization as the residual operation.

## 4    MASKED ATTENTION HEAD

Spatial locality plays a crucial role in computer vision tasks. CNN models capture it using the sliding filter of shared weights, typically with a receptive field of 3×3, 5×5, or 7×7. In contrast to CNN models, the locality is not explicitly introduced in the transformer structure. With attention masks, we can explicitly insert locality into self-attention modules without introducing any extra parameter or computation. The key idea is to apply a mask on the all-to-all attention products (i.e. $QK^T$) to reinforce the weights from the closest neighbors by aggregating information only from tokens selected by the mask.

Figure 1: Attention mask with depth of 1 (orange) and 2 (green).

Figure 1 illustrates an example of our proposed masking scheme. The orange box shows a $3 \times 3$ mask, where only the direct neighboring patches are selected. Specifically, Patch 16 only gathers information from the closest neighbors of Patch 1, 2, 3, 15, 17, 29, 30, and 31, and ignores the rest of patches. This is different from the typical all-to-all attention module, where Patch 16 attends to all 0-195 patches. We can easily expand the depth of the attention mask beyond the closest neighbors, to second-level neighbors (green box in Figure 1) or more. Note that the class token (and distillation token) still attends to all the patches to collect and spread information during forward and backward passes. Since each attention product selected by the mask is calculated by $Q$ and $K$, the masked attention head also retains the content-based locality information.

The attention mask is added before Softmax, regulating the distribution of attention maps to focus more on the closest neighbors:

$$Masked\ Attention(K, Q, V) = Softmax\left(\frac{M \odot QK^T}{\sqrt{d}}\right) V \qquad (2)$$

where $M \in \{0, 1\}^{(N+1) \times (N+1)}$ is a binary attention mask, encoding the spatial locality into the attention head by passing through only the weights from close neighbors and setting the rest to zero.

Note it's important to add the mask before Softmax because it allows the model to learn the importance of the locality flexibly. More precisely, unselected patches appear as $e^0 = 1$ in the numerator of the softmax operation. Thus, if the attention product result of the closest neighbors is meaningfully larger than zero (i.e. $M \odot QK^T \gg 0$), it suggests that local information dominates. However, if those results are negative or close to zero, it implies that local information is insignificant and global information is more important. *Therefore, inserting masks before the softmax operation allows models to enforce locality or bypass it.*

### 4.1    ATTENTION LOCALITY SCORE (ALS)

The softmax operation transfers the results of $QK^T$ into the probability space. As a result, each row of the attention map ($A$) sums up to 1. For each patch, the probability of focusing on local neighbors equals the sum of their neighbors' attention map weights. If this number approaches one, the local information is crucial; whereas if it is close to zero, it means global information matters more. For

Patch $n$, we define $ALS_n$ as the *Attention Locality Score (ALS)*. The mean of all $N$ patches is the *ALS* for each attention head, as follow:

$$ALS = \frac{\sum_n ALS_n}{N}, \text{ where } ALS_n = \sum_i^{N+1} (M \odot A)_{n,i} \tag{3}$$

where $M$ is the same attention mask defined earlier, and $A = Softmax\left(M \odot QK^T/\sqrt{d}\right)$ is the attention map, or $A = Softmax\left(QK^T/\sqrt{d}\right)$ for unmasked heads, and $n, i$ denotes the row and column index of the attention map respectively. We later use the ALS metric to get some insights about the locality behavior of different attention heads at various layers in our models.

## 4.2 MASKING STRATEGY

With total $H$ attention heads, $h'$ number of attention heads can be allocated to focus on local information. The rest of the unaltered attention heads $(H - h')$ capture global dependencies. Therefore, we can extract the local information through masked attention heads and global information through original attention heads at the same time, as shown in Figure 2.

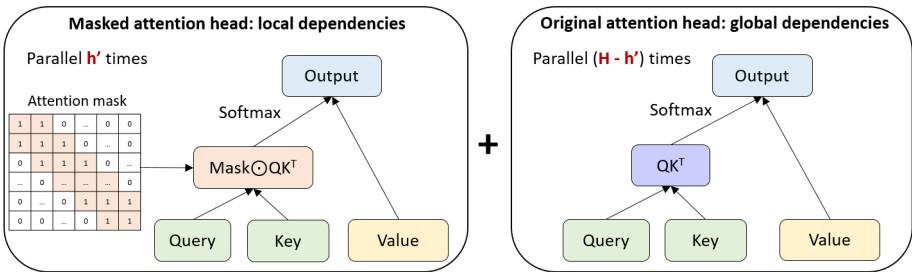

Figure 2: Masked attention heads and original attention heads for local and global dependencies.

Besides the depth of the attention mask, the number of masked attention heads in MHA and the position (which layer to insert mask) are also hyper-parameters. Though masks encode locality into the attention heads, the regularization from masking (pruning attention map) can also limit the learning capacity. Consequently, where to add attention masks requires careful consideration. The complexity to search for the best masking strategy is up to $2^{36}$ for a 12-layer model with 3 heads. Thus for a quick empirical search to obtain a decent mask placement within 2-3 training iterations, we leverage the insights from ALS study in Section 5.4 and follow a masking strategy below:

1. **Initialization**: add mask to only one attention head (Head 0) for all layers to explicitly introduce locality in every layer (forming a local path from the beginning to the end)
2. **Assignment**: compute $ALS^{0,l}$ for every layer $l$
   (a) If $ALS^{0,l}$ is close to 1: add masks to all attention heads of layer $l$, proceed to 3
   (b) If $ALS^{0,l}$ is close to 0.5: keep the mask, add no more
   (c) If $ALS^{0,l}$ is close to 0: remove the mask
3. **Calibration**: following 2.a), compute $ALS^{h,l}$ for each head and remove the mask if $ALS^{h,l}$ is close to 0

Note that optimal masking is a large space as opposed to a single unique placement. The model is capable of adjusting to different masking schemes flexibly. Step 2 (a) relies on the observations that early layers (typically with ALS close to 1) focus heavily on local features and thus more masked heads are beneficial. Step 3 will remove the redundant mask accordingly.

## 4.3 SOFT MASKING

To alleviate the complicated searching problem and automatically learn the mask placement, we introduce a learnable scale factor $\alpha \in (0, 1)$ for each attention head. We name this approach Soft

Masking, which replaces the zeros in the original masks with scale factors and keeps the ones. The binary masking approach is named Hard Masking.

$$M_{n,j} = \begin{cases} 1 & \text{if j is n's close neighbor} \\ 0 & \text{if j is not n's close neighbor} \end{cases} \text{ vs. } M'_{n,j} = \begin{cases} 1 & \text{if j is n's close neighbor} \\ \alpha & \text{if j is not n's close neighbor} \end{cases} \quad (4)$$

where $M_{n,j}$, $M'_{n,j}$ is the n-th row and j-th column element in the hard and soft mask respectively.

This scale factor penalizes the attention weights from non-neighboring patches. It allows non-local patch tokens to contribute differently. When $\alpha$ approaches 0, this attention head focuses more on local information. Otherwise when it's close to 1, the attention head attends global information. As a result, each attention head at every layer is able to flexibly determine the importance of locality. Note, this introduces negligible extra parameters during training. For example, it only introduces 36 extra parameters for a 12-layer MaiT tiny model with 3 heads.

## 5 EXPERIMENTS

In this section, we report the results for MaiT, implemented by applying attention masks on top of DeiT. We evaluate the impact of different hyper parameters related to the masking scheme including the mask depth, number of masked heads per layer, and number of transformer layers. We also analyze the importance of locality in the depth direction and the performance impact of masks for deep transformers. We merely focus on models with fewer than 50M parameters, which are more applicable to embedded systems and mobile devices.

### 5.1 SETUP

We follow the same training procedure as DeiT, unless specified otherwise. The implementation is based on the Timm library (Wightman, 2019). Models are trained with ILSVRC-2012 ImageNet dataset (Deng et al., 2009) with the default batch size of 1024 on 4 GPUs for 300 epochs for 12-layer models and 400 epochs for models with more than 12 layers. The model parameters we adopted is summarized in Table 1.

Table 1: Details of MaiT model parameters

| Model | Extra tiny (XT) | Tiny (T) | Extra small (XS) | Small (S) |
|---|---|---|---|---|
| Hidden dim. | 144 | 192 | 288 | 384 |
| Heads | 3 | 3 | 3 | 6 |

### 5.2 IMPACT OF MASK DEPTH

Intuitively, the field of view is larger with an attention mask of larger depth, to gather broader neighborhood information. However, in the extreme case with mask depth of $N+1$ (standard self-attention), all the patch token information is merged and positional information is lost. Nevertheless, considering the pruning effect from the mask, a smaller depth for the attention mask potentially translates to fewer FLOPs, which might be useful for some hardware devices such as embedded CPUs. For example, with the $3\times3$ mask, we only compute 9 out of all $(N+1)$ attention weights for each token. When $N=196$, this saves 95% of attention map computation.

To study the impact of the mask depth, we mask only one attention head for every MHA module, with the field of view as $3\times3$, $3\times5$, and $5\times5$ on tiny and small DeiT models. The results are summarized in Table 2. There is 0.1 - 0.3 % improvement compared to DeiT small. Various mask sizes only lead to marginal differences in accuracy. This is probably because each patch (original 16x16 pixels) is a processed sub-image and already contains some locality information, and thus further away patch tokens barely provide much extra information. Therefore, 3x3 mask is the default setting for MaiT in the subsequent sections.

In contrast to DeiT, whose accuracy saturates beyond 300 epochs as reported in (Touvron et al., 2021a), MaiT-small sees another 1.1% improvement when trained for 400 epochs. Accordingly, we

Table 2: Top-1 accuracy on ImageNet for DeiT and MaiT (one masked head for all 12 layers) with various attention mask depth. MaiT$^+$ adopts soft masking and is trained for 400 epochs.

| Model | DeiT | MaiT(3x3) | MaiT(3x5) | MaiT(5x5) | MaiT$^+$(3x3) |
|-------|------|-----------|-----------|-----------|---------------|
| Tiny | 72.2 | 72.6(+0.4) | 72.2(-) | 72.1(-0.1) | 74.7(+2.5) |
| Small | 79.8 | 79.9(+0.1) | 80.1(+0.3) | 80.0(+0.2) | 80.9(+1.1) |

suspect a higher learning rate and lower drop path rate are beneficial for MaiT-tiny since attention masks already provide some regulation during training. Changing the batch size from 1024 to 3072 only results in small gain of 0.4% for MaiT-tiny with 3×3 mask. In comparison, adopting 0 drop path rate and 2× learning rate, we obtain 2.5% improvement over DeiT-T.

## 5.3 IMPACT OF MASKING STRATEGY

We compare the performances of 24-layer MaiT models with the following three representative masking schemes (Sch.), to quantify the impact of the masking strategy. Sch.1 is a naive way to combine local and global information at every layer. Sch.2 results from the masking strategy in Section 4.2. Sch.3 is based on the soft masking.

**Sch.1**: one masked head for all 24 layers

**Sch.2**: Masked heads for Layer 0-7 is $H$-1, Layer 9-19 is one, and Layer 20-23 is none

**Sch.3**: Soft masking for Layer 0-20, no mask for Layer 21-23

Table 3: Top-1 accuracy on ImageNet for 24-layer DeiT and MaiT with different masking schemes. Sch.1 adopts only one masked head. Sch.2 applies a mixed masking and Sch.3 uses soft masking.

| | DeiT | MaiT(Sch.1) | MaiT(Sch.2) | MaiT(Sch.3) |
|--|------|-------------|-------------|-------------|
| Tiny | 78.3 | 79.1 | 79.1 | 79.3 |
| Extra Small | - | 81.0 | 81.4 | 81.3 |

As shown in Table 3, top-1 accuracy improves by 0.8% simply through adopting one masked head for MaiT-T. Another 0.4% improvement is achieved with mixed masking for MaiT-XS. Soft masking further increases the accuracy by 0.2% for MaiT-T. Note that the initialization of the scale factor for MaiT-XS is subject to further tuning.

Empirically we find Sch.2 yields relative good performances by masking heavily in the first one-third of layers to extract local features and only one mask for the middle layers. Note that there is a tradeoff between computation complexity and accuracy with hard masking and soft masking. Hard masking can translate to linear increase for attention map computation with respect to the number of tokens $N$, or $O(N)$, if hardware can leverage the structured sparsity. This is especially beneficial with high resolution images with a large number of tokens. The computation complexity remains quadratic as $O(N^2)$ for soft masking.

## 5.4 IMPORTANCE OF SPATIAL LOCALITY

Attention Locality Score (ALS) provides a quantitative metric to study the importance of spatial locality for each head across all layers. Figure 3 a) illustrates the result of a 24-layer MaiT-Tiny with one masked head. ALS of the masked head is significantly higher than other heads in Layer 0-22. This indicates the model learns to focus more on local information with the masked attention head. ALS gradually declines from around 90% for the first 8 layers down to around 60% and below 5% for the last layer. Note ALS of the last layer is below 0.0051 ($1/(N+1) = 1/197$), an average score if the attention weights are distributed uniformly. This suggests that spatial locality is largely ignored in the last layer where global information is more important.

Similarly, when learning the masking automatically through soft masking (Figure3 b)), the first 8 layers show over 0.9 ALS for multiple heads and layers. ALS gradually decreases in the middle layer with at most one head focusing heavily on local features. ALS is below 0.3 for the last 10 layers, where all heads tend to global information.

Overall, the spatial locality is more important in the early layers, and its importance gradually declines in deeper layer, which is consistent with d'Ascoli et al. (2021), Raghu et al. (2021)

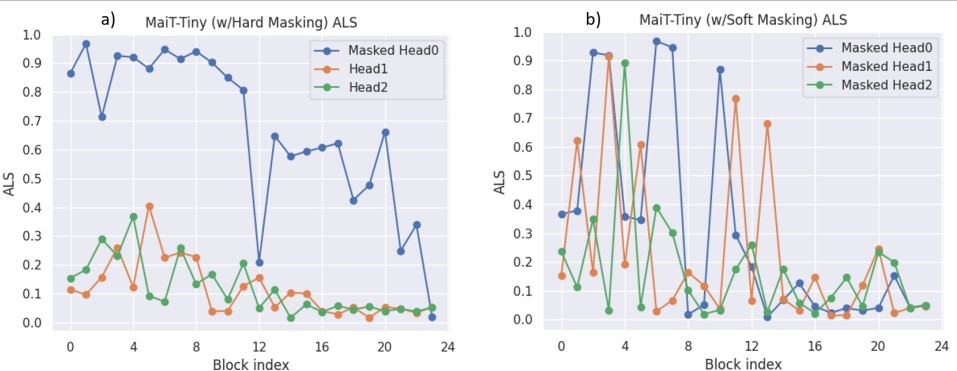

Figure 3: Attention Locality Scores (ALS) for MaiT-Tiny: a) one masked head for all 24 layers; b) soft masking for all 24 layers

## 5.5 DEEPER TRANSFORMERS

Another difficulty is to train deeper models because naively stacking transformer layers fails to deliver the expected performance gain as shown in Touvron et al. (2021b). One reason is attention collapse, i.e. the attention maps are more alike among deeper layers (Zhou et al., 2021). We find mixed masking is able to break the structural repetition in deep transformers and promotes diversity in later layers.

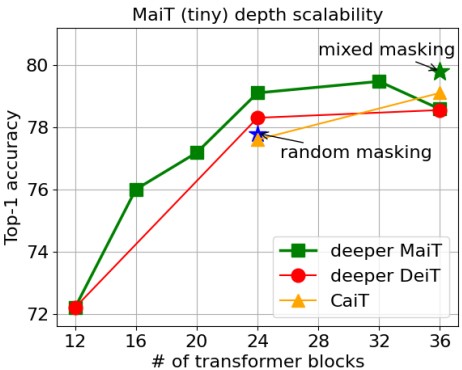

Figure 4: Top-1 accuracy on ImageNet for DeiT (red), MaiT (green), CaiT (yellow) and randomly masked DeiT (blue) with various numbers of transformer blocks. Green square MaiT applies one mask heads for all layers while green star MaiT masks all three heads for the first 12 layers and only one masked attention head for the rest 24 layers.

Performances of DeiT start to saturate after reaching certain depth. For example, increasing the number layers from 24 to 36, top-1 accuracy only increases by 0.25% for DeiT-tiny (Figure 4). In contrast, MaiT with mixed masking boosts the accuracy to 79.8%, extending the performance gain with deeper transformers. Interestingly, we noticed that for 24-layer model, CaiT is even worse than naively stacked DeiT-24, suggesting that the hyper-parameters for 24-layer CaiT-tiny are probably not optimal.

To rule out the pure pruning effects on attention maps from masking, we also apply a random mask of the same drop-out rate to one head of all 24 layers. The randomly masked model leads to 0.5% accuracy drop compared to 24-layer DeiT. This provides further proof that the performance gain with attention masks is indeed from better spatial locality aggregation.

To better understand the impact of attention masks in deep MaiT, we compare the cross layer similarity heat map among 36 layers of DeiT, MaiT (one masked head), and MaiT (mixed masking) in Figure 7, Figure 8, and Figure 9 respectively. Effective masking lowers the cross-layer similarity and thus improves scalability of MaiT.

## 5.6 COMPARISON WITH OTHER STATE OF THE ART VISION TRANSFORMERS

Attention mask can be easily combined with other optimization techniques. For deep transformers, we adopt the LayerScale (Touvron et al., 2021b) to MaiT as MaiT*. As shown in Table 4, MaiT/MaiT* obtain competitive performances across different model sizes. Specifically, MaiT-T outperforms CaiT-T by 1.7%. MaiT-S* achieves 0.2% higher accuracy than CaiT-S with less parameters/FLOPs and simpler structure, which translates to higher throughput. Moreover, simply adding the attention mask to CaiT leads to 0.5% and 0.2% higher accuracy for tiny and smaller models respectively.

Table 4: Top-1 accuracy for deep MaiT and some SOTA vision transformer models. *Added Layer-Scale on MaiT.

| Model | Layers | Params(M) | GFLOPs | Top-1 Acc. |
|---|---|---|---|---|
| LocalViT-T (Li et al., 2021b) | 12 | 5.9 | 1.3 | 74.8 |
| **MaiT-XT*** | 24 | 6.3 | 1.5 | **75.5** |
| PVT-T (Wang et al., 2021a) | 8 | 13.2 | 1.9 | 75.1 |
| CaiT-T (Touvron et al., 2021b) | 26 | 12 | 2.5 | 77.6 |
| **Masked CaiT** | 26 | 12 | 2.5 | **78.1** |
| **MaiT-T** | 24 | 11 | 2.5 | **79.3** |
| PVT-S (Wang et al., 2021a) | 15 | 24.5 | 3.8 | 79.8 |
| LocalViT-S (Li et al., 2021b) | 12 | 22.4 | 4.6 | 80.8 |
| Swin-T (Liu et al., 2021) | 12 | 29 | 4.5 | 81.3 |
| T2T-ViT-14 (Yuan et al., 2021b) | 16 | 21.5 | 5.2 | 81.5 |
| Twins-SVT-S (Chu et al., 2021a) | 18 | 24 | 2.8 | 81.7 |
| CaiT-XS (Touvron et al., 2021b) | 26 | 26.6 | 5.4 | 81.8 |
| **MaiT-XS*** | 24 | 24.5 | 5.3 | **81.8** |
| PVT-M (Wang et al., 2021a) | 27 | 44.2 | 6.7 | 81.2 |
| T2T-ViT-24 (Yuan et al., 2021b) | 26 | 64.1 | 14.1 | 82.3 |
| CaiT-S (Touvron et al., 2021b) | 26 | 46.9 | 9.4 | 82.7 |
| **Masked CaiT-S** | 26 | 46.9 | 9.4 | **82.9** |
| **MaiT-S*** | 24 | 43.3 | 9.1 | **82.9** |
| Swin-S (Liu et al., 2021) | 24 | 50 | 8.7 | 83.0 |
| Twins-SVT-B (Chu et al., 2021a) | 24 | 56 | 8.3 | 83.2 |

## 6 CONCLUSION

In this work, we incorporate spatial locality into vision transformers by inserting masks to attention heads. Masked heads are able to focus on local information and liberate other unmasked heads to extract global information more effectively. We also introduce attention locality score to guide the mask search and rapidly evaluate various masking schemes. Attention masking is a simple and effective technique, especially for small and deep transformer models. We observe that attention masks also serve as a regularizer to guide the training of attention maps for deeper transformers. Moreover, it is promising to further boost performances by combining this attention mask technique with other optimization approaches.

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

# A APPENDIX

## A.1 HYPER PARAMETERS FOR MAIT

Table 5: Hyper parameters for 24-layer MaiT in Table 4. Drop rate refers to linear stochastic depth drop rate, the same as in Touvron et al. (2021a). *denotes the constant drop out for all layers as in Touvron et al. (2021b)

| Model | Drop rate | Batch size | Learning rate |
|---|---|---|---|
| MaiT-XT | 0.0 | 1024 | 0.001×batchsize/512 |
| MaiT-T | 0.05 | 1024 | 0.001×batchsize/512 |
| MaiT-XS | 0.1 | 1024 | 0.0005×batchsize/512 |
| MaiT-S | 0.1* | 1024 | 0.0005×batchsize/512 |

## A.2 ALS OF DEIT AND DEEP MAIT

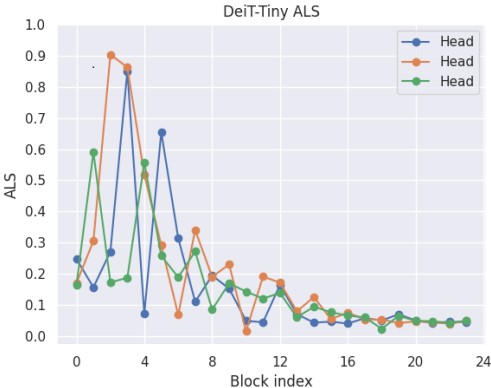

Figure 5: ALS for 24-layer DeiT tiny.

When trained without restriction from attention masks, DeiT emphasizes the locality in the first 8 layers with over 0.8 ALS for some heads, as shown in Figure 5. In comparison, in Figure 3, MaiT utilizes masked Head 0 to extract local information and ALS is lower for unmasked Head 1 and Head 2.

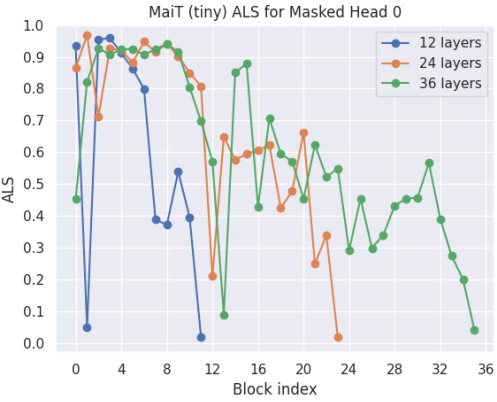

Figure 6: ALS of deeper MaiT-tiny with 12, 24 and 36 layers.

When the number of transformer blocks increases, the importance of locality expands to deeper layers as shown in Figure 6. For the masked Head 0 in MaiT-tiny, 12-layer model shows more than 0.8 ALS for 6 layers out of the first 7 layers, the following 4 layers give a ALS around 0.5, and the ALS of the last layer declines below 0.05. Likewise, for 24-layer and 36 layer models, 10 layers out of the first 12 layers show ALS higher than 0.8, and decreases to 0.5 or 0.6 in the following layers before the last layer.

### A.3 CROSS-LAYER SIMILARITY

To evaluate the impact of attention masks on the diversity of the attention maps across all layers, we apply a cross-layer similarity metric similarly defined in DeepViT (Zhou et al., 2021):

$$M_{h,t}^{i,j} = \frac{\mathbf{A}_{h,t,:}^{i} \mathbf{A}_{h,t,:}^{j\top}}{\|\mathbf{A}_{h,t,:}^{i}\| \|\mathbf{A}_{h,t,:}^{i}\|} \tag{5}$$

where $M_{h,t}^{i,j}$ is the attention map cosine similarity between layer i and layer j for attention head h and token t. $\mathbf{A}_{h,t,:}$ measures the weight contribution for each token in input (Value) for output token T. Therefore, it reflects the similarity of attention maps on information aggregation across all T input tokens. For example, if $M_{h,t}^{i,j}$ reaches 1, output token t at head h attends to all N+1 tokens in Value with the same probability for both layer i and layer j.

DeiT shows the highest cross-layer similarity at the last 8 stages, with an average of 0.75 (Figure 7). This homogeneity in attention maps at the late stage limits the model learning capability. Adding mask to one head for all 36 layers does not alleviate this problem since the same structure is repeated 36 times in the model. As a result, the average cross-layer similarity for the last 8 layers with unmasked heads is also 0.75 (Figure 8). Alternatively, if we mask all heads in the first 8 layers, they are naturally different from the rest of the partially masked layers, and the same structure is only repeated 8 or 24 times. As shown in Figure 9, cross-layer similarity between the first 8 layers and the following 24 layer are mostly well below 0.3 for Head 1 and 2. Additionally, a mixed masking scheme lower the cross-layer similarity among the last 8 layers by 0.1 to 0.65.

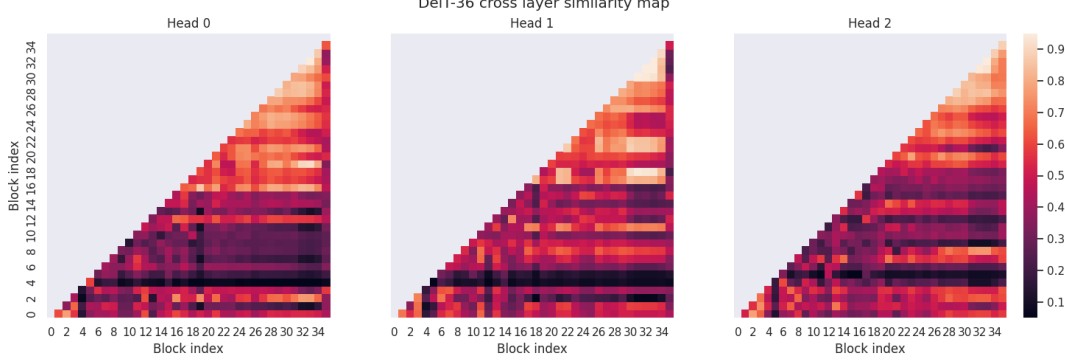

Figure 7: Attention map similarity across 36 layers for DeiT-tiny

In Figure 7, 36-layer DeiT-tiny shows the highest similarity at the last 8 stages across all three heads, with an average similarity of 0.75, whereas the similarity is quite small among the first 16 layers and between the first 16 layers and the last 20 layers. This is agreement with Zhou et al. (2021), indicating an attention collapse at later stage of deep transformers.

In comparison, for MaiT with one masked head across all layers, the masked attention Head 0 shows a more uniform distribution with a lower peak intensity as compared to the the non-masked or partially masked (in early layers) Head 1 and Head 2 (Figure 8. For unmasked Head 1 and Head 2, later stage especially the last 8 layers show highest similarity among each other, the average of them is also 0.75. This suggests one masked head for all transformer blocks is not as effective in 36-layer model as in 24-layer model. This indicates repeating the same structure for 36 times in the

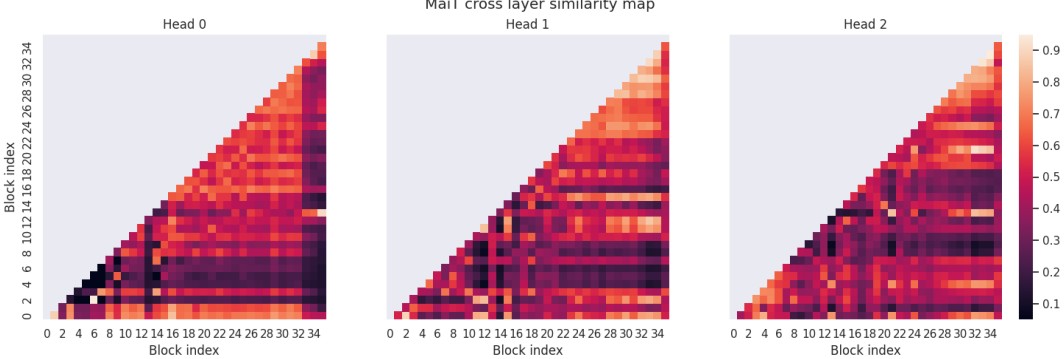

Figure 8: Attention map similarity across 36 layers for MaiT-tiny

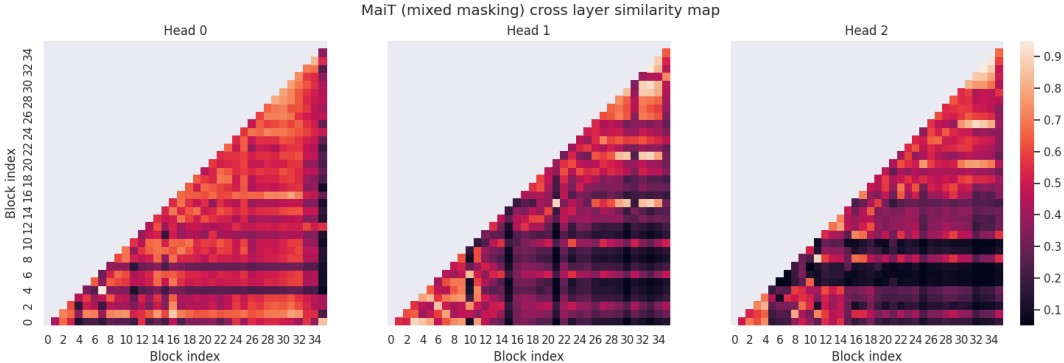

Figure 9: Attention map similarity across 36 layers for MaiT-tiny with mixed masking scheme

depth direction is causing the attention collapse. The diversity among the heads within the same layer doesn't alleviate this issue.

