# OpenReview forum: "MaiT: integrating spatial locality into image transformers with attention masks "
_ICLR.cc/2022/Conference — ICLR 2022 Submitted_

### Official Review · Reviewer_kZat · 2021-10-29

**Correctness:** 3
**Technical Novelty And Significance:** 3
**Empirical Novelty And Significance:** 3
**Recommendation:** 3
**Confidence:** 5

**Main Review:**

Pros:

- The idea of this paper is interesting. Though there have been many papers discussing the importance of introducing locality into vision transformers, this paper starts from using a mask to improve the self-attention.

- Experiments on DeiT show that the proposed approach performs better than the baselines, especially when two masked heads are used as shown in Table 2.

- Analysis is also given to demonstrate why involving attention mask could benefit training deeper vision transformers.

Cons:

- The arguement in Page 4 seems not correct. The authors claim that "the locality is not intrinsically encoded in the transformer structure."
In my view, it should be the locality is not explicitly introduced in transformers. As shown in the DeepViT paper, lower transformer blocks can indeed capture local information. I suggest the authors to reorganize the presentation here.

- The mathematic symbols should be italic. I think the authors should take a correct attitude towards paper writing.

- Experiments are insufficient. After the DeiT paper, there are a large number of papers working on vision transformers. In my view, taking the DeiT paper as baseline is already out-of-date. It would be better to show results on stronger baselines, such as CaiT, LV-ViT, which do not change the self-attention itself. If based on a stronger baseline, the proposed approach can still perform well, I think it should be a significant contribution.

- The improvement over the baselines is also not significant. Most previous work, like CaiT, LV-ViT, T2T-ViT, improve the DeiT model by more than 1% in terms of Top-1 accuracy. However, the improvement shown in the paper is lower.

- I do think a comparison with other methods for vision transformers should be added. I cannot find any tables showing such a comparison.

**Summary Of The Paper:**

This paper attempts to improve the classic vision transformers, or specifically the DeiT model by introducing the Masked Attention Head. Instead of focusing on aggregating global information in the original self-attention heads, this paper introduce local information via the proposed Masked Attention Head into self-attention. Experiments on ImageNet show that the proposed approach does matter for lifting the model performance of the original DeiT model.

**Summary Of The Review:**

Based on the concerns listed above, I think at this moment this paper does not reach the status for publication in ICLR.

---

> ### Author Response · Authors · 2021-11-23
> **Responses to Reviewer kZat**
>
> 1. This is a good point. We have updated it in the revision.
>
> 2. Mathematical symbols have been updated to be italic. We thank the reviewer for helping us get it right.
>
> 3. **Comparison with other vision transformers**:
> * We have updated Table 4 to include some other SOTA vision transformers. MaiT outperforms DeiT, PVT, LocalViT, T2T-ViT, and CaiT by a significant margin up to 4.2%. MaiT achieves comparable performances with Swin and Twins with a simpler structure and much fewer parameters.
> * LV-ViT adopts the label smoothing and introduces the extra training objective similar to knowledge distillation. Therefore a direct comparison between LV-ViT and MaiT seems unfair. The optimization technique used by LV-ViT is orthogonal to MaiT and combining them can potentially lead to higher performance gain.
>
> 4. The improvement from the binary attention mask is almost free, since it introduces no extra parameters/FLOPs or any structural change. In contrast, CaiT and T2T-ViT change model structure, and LV-ViT adds another training objective.

---

### Official Review · Reviewer_71Sc · 2021-10-29

**Correctness:** 2
**Technical Novelty And Significance:** 2
**Empirical Novelty And Significance:** 1
**Recommendation:** 1
**Confidence:** 5

**Main Review:**

**Strength**:

Incorporating local information efficiently into global self-attention is a promising research direction.

**Weakness**:

1. Writing is very poor, which makes it hard for me to understand what the authors want to express.

2. Notations are frustratingly messy and I can only list a small part of mistakes here. For example, at the beginning of Section 3, this paper fixes the size of input images, patch size, hidden dimensions, etc. The authors should define symbols clearly and specify the configurations in the experiments.
In Eq. (1), what are the definitions of Q, K, V?  In Eq. (3), what’s the meaning of index i? I guess it does not represent the layer index here? But in Section 4.2, i indicates the layer index.

3. The masking strategy, which is the core component of the paper, has serious technical problems.
    - The paper proposes a heuristic strategy to add masks. However, in Transformers, different heads actually encode distinct patterns (e.g., [A1]). However, in this paper, all the heads assume to share the same mask which is not reasonable. For 2 (a), why add masks to all attention heads? For 2(b), if the score is around 0.5, why only add a mask to the 0-th head? More explanations are needed.
    - The paper analyzes the search space but no search algorithm is proposed. To automatically search the masking strategy is important and it is not OK to leave it for future work.

4. The advantage of the proposed masked attention over the existing efficient Transformers with local-global self-attention (e.g., PVT [Wang el al., 2021a], Swin [Liu et al., 2021a] and Twins [Chu et al., 2021a]) is unclear.  In addition, the paper should compare with these methods.

5. The network configurations for MaiT-XT, Mai-T, MaiT-XS, MaiT-S are not clearly explained.

6. The paper argues that the proposed masked attention heads can facilitate the convergence but I failed to find any training and validation loss curves to justify this statement.

7. In Figure 1, why does the green mask correspond to the 72-th patch? In Figure 2, what does ‘-’ mean in the visualized attention mask?

Overall, I think the submission is not complete at the current stage and needs to be significantly improved.

**References**:

[A1]: On the Relationship between Self-Attention and Convolutional Layers, in ICLR 2020

**Summary Of The Paper:**

This paper proposes to introduce attention masks to incorporate spatial locality into part of self-attention heads, while keeping the remaining heads to capture global dependencies. Experiments show improved accuracy on ImageNet in comparison with some baselines.

**Summary Of The Review:**

The advantage of the proposed masked attention over the existing literature is not properly discussed. In addition, paper writting is very bad, with substansive confusing descriptions and notations.

---

> ### Author Response · Authors · 2021-11-23
> **Responses to Reviewer 71Sc**
>
> 1. We have fixed some typos and notation errors. We’ve also added Section 4.3 and 5.3 and updated the manuscript with some clarifications.
> 2. We thank the reviewer for pointing out the notation errors. We have updated the symbols accordingly. Q, K, V are query, key and value, which is clarified in the text above equation (1). The index i in Eq. (3) denotes the i-th row of the attention map with the clarification added below Eq. (3). We have updated the symbol in section 4.2 to avoid confusion.
> 3. **masking strategy**
> *Clarification*: This empirical mask search strategy aims for a good mask placement in only 2-3 training iterations when compute resources are limited. It also relies on the observations that early layers tend to focus more on local features and thus more attention heads are likely beneficial.
> * Step 1: initialization - one masked head for all layer to explicitly introduce locality in every layer and keep a local path from the beginning to the end. This also provides insights on the importance of locality along the depth direction.
> * Step 2: assignment - more masked head for layers with high ALS
> * Step 3: calibration - update the mask and remove redundant mask
>
> *Automatic search*: We added the soft masking in Section 4.3 with new experiment results in Section 5.3. Soft masking introduces a trainable scale factor (in the range of 0 to 1) for each attention head, which learns to penalize the attention weights from non-local patch tokens flexibly. This further boosts performances.
>
> 4. Comparison with other vision transformers:
> * MaiT applies the attention mask to enhance the multi-head attention module, which does not require any network structural change or adding any parameter/FLOP. Other vision transformers introduce the locality with some specific network structures, such as a hierarchical structure of progressive shrinking pyramid structure for PVT, shifted windows for Swin, or interleaved locally-grouped self-attention and global sub-sample attention layers for Twins.
> * We have updated Table 4 to include some other SOTA vision transformers. MaiT outperforms DeiT, PVT, LocalViT, T2T-ViT, and CaiT by a significant margin up to 4.2%. MaiT achieves comparable performances with Swin and Twins with a simpler structure and much fewer parameters.
>
> 5. Network configurations: Network configuration is specified in section 5.1 Table 1. MaiT with 12 layers shares the same structure with DeiT for fair comparison. Deep MaiT increases the number layers while maintaining the same network structure.
>
> 6. By facilitating the convergence, we refer to the fact that without masking, 24 layer small DeiT model fails to converge as shown in the CaiT paper.
>
> 7. We have fixed the typo in Figure 1 as it should be Patch 36. In Figure 2, it annotates the number of global heads as H - h’. The figure is updated to make it clearer.

---

### Official Review · Reviewer_PxTP · 2021-11-02

**Correctness:** 4
**Technical Novelty And Significance:** 2
**Empirical Novelty And Significance:** Not applicable
**Recommendation:** 5
**Confidence:** 5

**Main Review:**

Pros:
1. The paper is well-written.
2. The experimental results shows that improvement of the proposed method compared with the baselines.

Cons:
1. The idea of this paper is closely related to Swin Transformer. Swin transformers bring locality into network architecture by the window attention, which could save computation significantly. Yet, using attention masks does not have this additional benefit.
2. Thus, whether the proposed method could be applied to other transformers with locality in the attention module is questionable.
3. In the experiments, other transformers with locality mechanism (Swin transformer, LocalViT, T2T-ViT) should be compared.


**Summary Of The Paper:**

This paper proposes to bring locality into the attention module of vision transformers. This locality mechanism is brought by the introduced attention masks. Basically, the attention masks are binary and is likely to restrict the attention to the local field of a token. The local attention mask results in a block matrix before the application of Softmax function. Tokens with zero mask values are given a constant attention value of 1. Thus, those tokens are still involved in the computation. It is claimed that the binary masks is able to keep the global connection when necessary.

**Summary Of The Review:**

The main concern of this paper is the usefulness of the locality mechanism due to the existence of Swin transformers which are more computationally efficient.

---

> ### Author Response · Authors · 2021-11-23
> **Responses to Reviewer PxTP**
>
> 1. **Comparison with Swin**: Both Swin and MaiT extract spatial locality from patches within a window, however they are different in the follow aspects:
> * Patches outside the window still contribute to attention weights for MaiT.
> * Within each layer, Swin uses larger **non-overlapping windows** (7x7) while MaiT applies smaller **overlapping windows** (3x3 mask)
> * To gather global information, Swin introduces shifted window across layers while MaiT uses unmasked attention head
> * MaiT is not constrained to any specific model structure.
>
> Regarding computation, Swin gradually reduces the number of tokens while increasing the channels with deeper layers. In the early layers, Swin requires more computation due to the smaller patch size of 4x4 instead of 16x16 in DeiT/MaiT. For 224x224 input resolution, Swin (with a more complicated structure) results in lower throughput compared to DeiT (simpler monolithic structure), as shown in Swin paper.
>
> Besides, MaiT introduces structured sparsity in the attention maps. If the hardware can leverage this static sparsity, the complexity of attention map computation reduces to O(N) from O(N^2). This would be significant saving with high resolution images.
>
> 2. **Applicability to other transformers** Attention-mask based approach is independent of many other optimization techniques. Combining them is promising to obtain further performance gain. Simply adding binary masks improves CaiT-T and CaiT-S by 0.5% and 0.1% respectively. This performance gain is almost free with no added parameters/FLOPs or any structural change.
>
> 3. **Comparison with other vision transformers** We have updated Table 4 to include the comparison with Swin, LocalViT, T2T-ViT, and others. MaiT outperforms LocalViT by 0.7-1.0%, and is 0.3-0.5% better than T2T-ViT at different model-size categories. MaiT-XS outperforms Swin-T by 0.5% with similar parameters and FLOPs. MaiT-S is 0.2% lower than Swin-S but with 6.7M less parameters.

---

### Official Review · Reviewer_wzwv · 2021-11-03

**Correctness:** 3
**Technical Novelty And Significance:** 2
**Empirical Novelty And Significance:** 2
**Recommendation:** 5
**Confidence:** 3

**Main Review:**

strengths:
- the paper proposes a masking strategy for attention mechanism, to incorporate the spatial locality into self-attention heads.
weaknesses :
- how is the proposed method differs from the multi-scale transformers or hierarchical vision transformers
-  novelty for masking strategy is minimal.
- Can the proposed method be extended to any kind of transformers.
- it hard to understand how the masking strategy addresses the spatial locality-based attention mechanism


**Summary Of The Paper:**

The paper proposes a method that increased DeiT performance by 1%, where the proposed method introduces attention to incorporate spatial locality into self-attention heads. By doing this paper claims that local dependencies are captured with masked attention heads along with global dependencies captured by original unmasked attention heads.

**Summary Of The Review:**

In summary the proposed method has few strengths and weaknesses, it would be easy to understand the effectiveness of the proposed method, if the paper provides experiments showing the improvements when masking strategy is introduced to other transformer networks

---

> ### Author Response · Authors · 2021-11-23
> **Responses to Reviewer wzwv**
>
> 1. **Comparison with multi-scale or hierarchical vision transformers**
> Attention mask based approach directly encodes the locality into multi-head attention modules. This introduces no structural changes and thus offers a wider applicability. Besides, MaiT is a monolithic model, which preserves a simpler structure and one-to-one token correspondence throughout all layers. This is beneficial for some unsupervised learning frameworks [A1] to learn good semantic correspondence, whereas hierarchical transformers lose this one-to-one token correspondence due to token merging along the depth direction.
>
> A1. Efficient self-supervised vision transformers for representation learning, arxiv 2021
>
> 2. **Novelty of the masking strategy**
> While preserving spatial locality is a well-known concept in computer vision models, our masking strategy offers the following novelties:
> * We Introduce attention locality score (ALS) to evaluate the importance of locality and guide the mask placement.
> * We propose the soft masking to apply a learnable scale factor for each head and learn the mask placement automatically.
>
> 3. **Applicability to any transformer** Attention-mask based approach is independent of many other optimization techniques. Combining them is promising to obtain further performance gain. Simply adding binary masks improves CaiT-T and CaiT-S by 0.5% and 0.1% respectively. This performance gain is almost free with no added parameters/FLOPs or any structural change.
>
> 4. **masking strategy** Masked heads focus on patches from closest neighbors and discount non-local patches.
> The empirical mask search strategy aims for a good mask placement in only 2-3 training iterations when compute resources are limited. It also relies on the observations that early layers tend to focus more on local features and thus more attention heads are likely beneficial.
> * Step 1: initialization - one masked head for all layer to explicitly introduce locality in every layer and keep a local path from the beginning to the end.
> This also provides insights on the importance of locality along the depth direction.
> * Step 2: assignment - more masked head for layers with high ALS
> * Step 3: calibration - update the mask and remove redundant mask

---

### Author Response · Authors · 2021-11-23
**Paper revision, key contributions and clarifications**

We thank all the reviewers for their valuable comments. We restate our key technical contributions and insights:
1. Attention masks integrate locality into the self-attention module without introducing any structural change or extra parameters/FLOPs.
2. We propose a quick empirical masking search strategy that leads to decent performances within 2-3 training iterations.
3. Alternatively we also introduce a soft masking with a trainable scale factor for each head to learn masking placement end-to-end automatically.
4. MaiT achieves competitive performances using a simpler model and fewer parameters compared to many SOTA vision transformers.

We have updated the manuscript with the following **key changes**:
1. **Masking strategy**: We added the soft masking in Section 4.3 with new experiment results in Section 5.3. Soft masking introduces a trainable scale factor (in the range of 0 to 1) for each attention head, which learns to penalize the attention weights from non-local patches flexibly. This further boosts performances.
2. **Comparison with other vision transformers**: We have updated Table 4 to include some other SOTA vision transformers. MaiT outperforms DeiT, PVT, LocalViT, T2T-ViT, and CaiT by a significant margin up to 4.2%. MaiT achieves comparable performances with Swin and Twins with a simpler structure and much fewer parameters.
3. **Applicability to other vision transformer models**: Due to limited time and compute resources, we have only tested it on CaiT. Mixed binary-mask improves the accuracy of CaiT-T and CaiT-S by 0.5% and 0.1% respectively at no cost of extra parameters/FLOPs.

We also highlight below some **key clarifications** to address some common concerns:
1. **Empirical binary-masking search**: The empirical mask search strategy aims for a good mask placement in only 2-3 training iterations when compute resources are limited. It also relies on the observations that early layers tend to focus more on local features and thus more attention heads are likely beneficial.
* Step 1: initialization - one masked head for all layer to explicitly introduce locality in every layer and keep a local path from the beginning to the end. This also provides insights on the importance of locality along the depth direction.
* Step 2: assignment - more masked head for layers with high ALS
* Step 3: calibration - update the mask and remove redundant masks

2. **Difference from other vision transformers**: MaiT applies the attention mask to enhance the multi-head attention module, which does not require any network structural change or adding any parameter/FLOP. This approach potentially has a wider applicability. Other vision transformers introduce locality with some specific network structures, such as a hierarchical structure of progressive shrinking pyramid structure for PVT, shifted windows for Swin, or interleaved locally-grouped self-attention and global sub-sample attention layers for Twins. LocalViT introduces depth-wise convolutions into the feed-forward network to better capture locality.

3. **Comparison with Swin**: Both Swin and MaiT extract spatial locality from patches within a window, however they are different in the follow aspects:
* Patches outside the window still contribute to attention weights for MaiT.
* Within each layer, Swin uses a larger non-overlapping window (7x7) while MaiT applies a smaller overlapping windows (3x3 mask).
* To gather global information, Swin introduces shifted window across layers while MaiT uses unmasked attention head within the same layer.
* MaiT is not constrained to any specific model structure.

Regarding computation, Swin gradually reduces the number of tokens while increasing the channels with deeper layers. In the early layers, Swin requires more computation due to the smaller patch size of 4x4 instead of 16x16 in DeiT/MaiT. For 224x224 input resolution, Swin (with a more complicated structure) results in lower throughput compared to DeiT (simpler monolithic structure), as shown in Swin paper.

Besides, MaiT introduces structured sparsity in the attention maps. If the hardware can leverage this static sparsity, the complexity of attention map computation reduces to O(N) from O(N^2). This would be significant saving with high resolution images.

---

### Decision · Program_Chairs · 2022-01-20

**Decision:**

Reject

**Comment:**

The paper presents a masking strategy to introduce the locality bias into the vision transformers. The experiments show the effectiveness of considering such inductive bias. The reviewers agreed on the importance of the research question and the simplicity of the algorithm. MaiT also has a straight-forward sparse attention extension that performs on the complexity of $O(n)$ rather than $O(n^2)$.

The reviewers also listed some common concerns of the paper:

(1) The novelty of such a masking approach is relatively low. I don't think the ALS or the soft masking adding too much contribution to that. Similar ideas have been explored in a number of papers.

(2) Reviewers also raise concerns about the experiments. Inductive biases often help more in small settings (fewer parameters and FGLOPs) and gain less in the large settings. When comparing with the STOA models, I think this is basically the trend shown in the paper as well. While I appreciate the authors’ efforts in including more comparisons, I have to say I really don’t think the performance gain is significant enough especially in the large settings. Needless to say that there are many other ways of encoding the same locality bias into the model.

Based on the reviewers' judgements and my own opinion, I therefore recommend rejection of this paper.